# Measuring Erlang-Based Scalability and Fault Tolerance on the Edge

**DOI:** 10.3390/s25154843

**Published:** 2025-08-06

**Authors:** Daniel Ferenczi, Gergely Ruda, Melinda Tóth

**Affiliations:** 1Faculty of Informatics, Eötvös Loránd University, 1117 Budapest, Hungary; tothmelinda@elte.hu; 2Evosoft Hungary Kft., 1117 Budapest, Hungary; gergely.ruda@evosoft.com

**Keywords:** IoT, Erlang, BEAM, AtomVM, fault tolerance

## Abstract

Embedded systems in IoT are expected to be run by reliable, resource-efficient software. Devices on the edge are typically required to communicate with central nodes, and in some setups with each other, constituting a distributed system. The Erlang language, favored for its constructs that support building fault-tolerant, distributed systems, offers solutions to these challenges. Its dynamic type system and higher-level abstractions enable fast development, while also featuring tools for building highly available and fault-tolerant applications. To study the viability of using Erlang in embedded systems, we analyze the solutions the language offers, contrasting them with the challenges of developing embedded systems, with a particular focus on resource use. We measure the footprint of the language’s constructs in executing tasks characteristic of end devices, such as gathering, processing and transmitting sensor data. We conduct our experiments with constructs and data of varying sizes to account for the diversity in software complexity of real-world applications. Our measured data can serve as a basis for future research, supporting the design of the software stack for embedded systems. Our results demonstrate that Erlang is an ideal technology for implementing software on embedded systems and a suitable candidate for developing a prototype for a real-world use case.

## 1. Introduction

The Internet of Things refers to a wide range of devices, including industrial devices, vehicles, and sensors, connected to larger systems that enable the analysis, presentation, and, in many cases, control of data. The use of such setups ranges from home automation and agriculture to the manufacturing industry, civil infrastructure management, and healthcare, to name a few examples. These systems use simple devices on their endpoints that typically perform designated tasks. Depending on the use case, they are often operated unmanned and deployed on a large scale. Consequently, their on-site maintenance is highly expensive and often unfeasible. Therefore, bug fixes are often impractical and costly to deploy.

However, discovering bugs during testing is more difficult for software running on end devices. As the final hardware often contains specialized components, development processes rely on using simulators and testbeds that may not capture all the features of the actual device. Nevertheless, using an actual device for development constrains iterative development and continuous testing.

In an IoT system, these devices are parts of a distributed system. They are expected to communicate with other, similarly configured devices, and with central components that work as aggregators or coordinators. Testing such connected systems presents its challenges, as we must account for delays and failures in communication, as well as the availability of other devices, among other challenges. These characteristics also make automated testing more difficult.

Overall, although bug fixes are challenging to deploy in IoT systems, finding bugs through conventional testing presents numerous challenges. Consequently, tools and techniques to avoid bugs and runtime errors are of high value. In general, software development practices for building IoT systems efficiently are the subject of research [1,2].

Aside from the correctness of software on the end devices of IoT systems, non-functional requirements include the overall power and memory consumption of the running application. IoT use cases may require on-site devices to operate on battery for months, making resource efficiency a central element of system design.

In our current work, we investigate how Erlang addresses the challenges typical of embedded development. We answer a variation of the hypothesis we presented in our previous work [3]: “Can industrial applications benefit of AtomVM as an Erlang runtime?”.

The main difference between our original and current hypotheses lies in targeting an exact runtime for the Erlang language. Our work focuses on AtomVM, an Erlang runtime developed specifically for the unique requirements of embedded use cases. We investigate our research question in detail via the following contributions:We design and implement a test environment (Section 3) for running a configurable number of concurrent Erlang processes. We try the distinct methods for fault tolerance that Erlang offers through variants of this system.To account for data transmission between IoT nodes, we also built a variant that transmits data through a wireless connection.We define Section 3.2 and evaluate Section 4 experiments to assess the cost of running fault-tolerant, supervised processes.We also compare the reliability and efficiency of wireless transmission using Erlang as opposed to an implementation based on C++.

As in practice, IoT end devices perform data acquisition, processing and wireless transmission. We designed our experiments to test Erlang-based implementations for each of these tasks.

These measurements serve as a basis for our ongoing research on using Erlang to build fault-tolerant software for the edge. We explore how the language’s features contribute to embedded development and how limited hardware constrains them. Understanding these limits will enable us to develop prototypes tailored to specific use cases in our future studies.

## 2. Background

In this section, we present the Erlang language and the challenges of IoT development. We focus on the challenges that we have found both significant and problems that the Erlang language helps to address. We also include a description of runtimes available for using Erlang in IoT setups. At the end of the section, we present related research on the application of Erlang and related technologies for embedded devices.

### 2.1. Erlang

Erlang is a functional programming language, initially developed by Ericsson in the 80s to address challenges in the telecommunications domain. Their goal was to create software that is highly scalable, fault-tolerant and allows for changing running software without downtime. These features also made the programming language popular in other domains. Notably, it is also used in banking, chat messaging, and online gaming. Our hypothesis on the benefits that Erlang can bring to IoT is based on these particular features.

Erlang applications achieve high availability by using lightweight processes that can even be spawned on a request basis. These are ordered into supervision structures capable of restarting crashed processes promptly. This approach helps to cleanly separate business logic from the failure handling of an application. Developers are encouraged to follow this pattern instead of using defensive programming. Erlang calls this pattern “let it crash”.

Erlang’s features are provided by its runtime, the BEAM virtual machine. Other languages were also invented that run on this virtual machine. Elixir, one such language, also shares Erlang’s distinguishing features, allowing developers to build scalable, fault-tolerant software.

These features are part of the language and are available for use without the addition of components to the default Erlang software stack. As a result, developers can work on scalable, fault-tolerant applications without needing to learn additional tools beyond the programming language or add components to their software bill of materials.

### 2.2. IoT Development Challenges

IoT setups often rely on end devices that operate in unmanned environments. If a system consists of several hundred or thousands of devices, its manual upgrade becomes unfeasible. By contrast, implementing over-the-air updates increases the device’s energy footprint [4] and also requires a secure infrastructure to ensure that only trusted software is deployed.

In some use cases, over-the-air updates may conflict with the design considerations of end devices, as these often require a low power footprint and are isolated from external networks, especially when updating their software.

Software updates can include both new features and bug fixes. We are particularly interested in software errors, as applying suitable software engineering techniques can reduce their number and effects before flashing the software on the final device. Choosing the appropriate development flow can be challenging, as the devices being developed interact with the physical environment, and it is difficult to design automated test methods that accurately simulate the conditions in which a device is deployed. Software patterns that isolate runtime errors and prevent the propagation of errors are therefore valuable. For example, systems can mitigate some errors by restarting only the failing modules of the software [5]. A practical design would allow data gathering, evaluation and transmission on a device’s working sensors to remain functional; hardware or software modules related to other sensors face runtime issues.

Runtime issues can also arise from unstable connectivity between nodes. The number of deployed units in an IoT system adds to the complexity of testing. A developer must account for large variations in the number of nodes that compose a system and the unreliability of their connections. Testing methods for distributed systems have been proposed. For example, the DieCast [6] project’s authors propose simulating scenarios with a large number of nodes. Such an approach has more limitations when validating IoT systems, as real nodes typically use wireless communication and both the channel and the device itself are subject to interference and other changes in the physical environment. Again, engineering methods that account for this complexity, such as separating business logic and fault tolerance, are beneficial.

Aside from the challenges related to runtime and testing, development is also harder on these devices. Although a simulator can be added to a continuous integration workflow, deployments to real devices are harder to automate. Actual work may also involve experimenting with different hardware components that require manual flashing of the devices. Additionally, unforeseen challenges may necessitate changes even to the hardware during the development phase. As a result, developers of embedded systems often work with slower feedback loops than those working on conventional applications hosted on servers.

Higher-level languages that are faster to develop in could make development faster. However, programming tools and, in general, resources for embedded development assume the use of conventional languages, such as C, C++, or Ada [7], for writing embedded software.

Finally, many use cases require low power draw from the end nodes. This places a constraint on the technology we choose to develop our software, as the final result must be energy-efficient. Some scenarios work with higher-powered devices, like the Raspberry Pi. Our focus, however, lies on using Erlang on low-powered microcontrollers. For comparison, while Raspberry Pi 3B boards can draw a current of up to 1340 mA under stress (https://www.raspberrypi.com/documentation/computers/raspberry-pi.html#power-supply, (accessed on 30 July 2025)), a microcontroller-based board like the Raspberry Pi Pico 2 draws about 12 mA while constantly transmitting characters to a serial console (https://datasheets.raspberrypi.com/pico/pico-2-datasheet.pdf, (accessed on 30 July 2025)). ESP32, STM32 and Raspberry Pi Pico microcontroller families typically also support sleep functions natively, further reducing power consumption. In the case of the ESP32S3-based WiFi LoRa 32 board from Heltec, this feature reduces consumption to 10 uA (https://resource.heltec.cn/download/WiFi_LoRa_32_V3/HTIT-WB32LA_V3.2.pdf, (accessed on 30 July 2025)).

### 2.3. Erlang’s Benefits in IoT Development

Erlang was designed to be a language for implementing fault-tolerant systems, initially in the telecommunications domain. The language designers’ goal was to build highly reliable telecom switches that could scale quickly and efficiently. They found that to this end, a new language is needed that supports lightweight processes, sharing information through the actor model [8], and includes features to manage process supervision and the development of common structures, such as servers and state machines. The language and its standard library were used in building Ericsson’s AXD301 switches, considered one of the most reliable systems [9].

Erlang’s advantage in building fault-tolerant, distributed systems helped its adoption outside the telecommunications domain, but still remained mostly in use in server-side applications, running on highly performant computers. As the language’s features are practical on lower-powered systems, in recent years, IoT-specific frameworks and runtimes have also been built: GRiSP [10], Nerves [11], and AtomVM [12]. While all three tools bring Erlang and other BEAM-related languages to IoT edge devices, their differences are important to note, as they serve different use cases.

GRiSP brings Erlang as close to the hardware as possible. It is only compatible with GRiSP boards and features a tight integration between the language runtime and the device’s real-time operating system, allowing hard real-time event handling.

Nerves provides a BEAM distribution for computers like the Raspberry Pi or BeagleBone and primarily brings its features to applications developed in Elixir. Both of these frameworks run on devices that are typically powered continuously, and their current releases do not work on low-powered devices.

In contrast, AtomVM runs on low-powered embedded devices. Erlang applications executed on AtomVM can consequently work on systems that run on batteries and do not require a continuous power supply. To achieve these low levels of power consumption, the developers of AtomVM reimplemented the BEAM virtual machine to contain only the features most important for embedded devices. Omitted features include bignums, an REPL and support for dynamically changing running code.

Erlang’s distribution protocol is also only included as a proof of concept on a feature branch. This feature enables the creation of distributed Erlang applications that run across multiple nodes, as well as the actions to perform when complete nodes become unavailable. Defining applications across multiple nodes enables the construction of true fault tolerance, which accounts for the general failure of endpoints. We find this feature an interesting candidate for IoT scenarios, and plan to research it in detail as it matures.

AtomVM’s general releases, however, include the Erlang features for building fault-tolerant applications and managing a high number of processes. It also contains pattern matching on binaries, a feature particularly useful when processing data read from sensors and communication channels of embedded devices.

In a study [13], Trinder found that, apart from natively offering these features, Erlang programs are also more concise compared to C++ implementations. The decrease in codebase size is attributed to the functional language constructs, the language’s programming style that encourages handling process crashes in a controlled manner rather than through defensive programming, and automatic memory management. Overall, the Erlang version consisted of shorter, more maintainable code that was more focused on the problem’s details rather than on communication and defensive handling of data.

Ray, Posnett and their colleagues researched the relationship between programming languages and code quality in their study [14]. Their work presents a connection between languages and defect categories. When comparing the impact languages have on defect categories, they found that although Erlang applications have more security related errors than C++, they have less memory, concurrency and general errors. Erlang’s small number of memory-related errors is of special note, as in this regard only Php projects had fewer defects.

These benefits come at the cost of memory. Embedded devices generally have a lower amount of memory available. In our study, we investigate how this limits process management in practice on an actual board.

Erlang’s dynamic type system also helps accelerate the development process. Dynamic typing typically makes it harder to find bugs during development, resulting in type errors at runtime. Erlang’s features for fault tolerance, however, mitigate this problem to a degree, making Erlang a preferred choice for creating highly robust software [15].

We believe that the features that help create resilient software can be particularly beneficial when developing embedded applications. As presented in Section 2.2, software developed for embedded devices is not only harder to test automatically but also expensive to patch. Built-in fault tolerance and a lower total codebase help writing code that is easier to maintain and review, and consequently can help reduce bugs in the system.

Erlang’s support for concurrency is built around the actor model [8]. Erlang processes do not communicate data through shared memory. They run in parallel and communicate data through sending messages to each other’s mailboxes. This design helps avoid all possible bugs rooted in invalid data being present in shared memory. Erlang’s processes have a tiny footprint and can be launched at high speed, supporting rapid changes in an application’s load. The programming model scales through processor cores and devices hosted in different locations. This pattern helps design software architectures for distributed systems common in IoT scenarios.

Although AtomVM has a reduced set of features compared to BEAM, those included still provide the main features that make Erlang systems resilient: lightweight, scalable processes, the actor model, and process supervision. We have summarized how Erlang’s features can help with challenges in developing embedded systems in Table 1. Naturally, constructs for fault tolerance, memory management, and features of higher-level functional languages all have a footprint in both memory and power consumption. In our research, we investigate these requirements in detail. We find both measurements valuable for future prototyping, as memory is typically limited on embedded devices and power requirement determines if the system is fit for a given purpose. We expect our results to support the evaluation of Erlang and AtomVM for specific use cases.

### 2.4. Related Work

Erlang’s and BEAM’s use in IoT and industry has been the subject of past research.

Xu, Michala and Trinder [16] investigated Elixir, another BEAM-based language in their CAEFL project. They developed a system for federated learning, where agents enhanced a global model with local information from their models. They also applied hot-code loads to change existing environments, define new ones, and add and remove agent sensors on a running system.

Takase [17], in his work, researched autonomous communication of end nodes running Nerves. To this end, his research team developed an Elixir library for Elixir applications running on Linux hosts, specifically for the Data Distribution Service (DDS) real-time networking middleware. They found Elixir’s high scalability beneficial for creating multiple subscribers and publishers. As DDS relies on native Linux features, they also implemented the required extensions to allow for their library to work on Nerves. They also made connections between different networks viable with the Zenoh protocol.

Nerves has also been investigated by Kikuchi [18], who used it for controlling a small water plant. The work’s goal was to introduce a technology suitable for both control and communication, as well as monitoring and logging. This is typically divided into factory automation (FA) and supervisory control and data acquisition (SCADA) subsystems. FA-level devices have limited features and are typically unsuitable for connecting to large networks that host business applications. Kikuchi was researching the use of a software stack on general-purpose hardware to merge both subsystems into a larger, distributed system. Initially, his team experimented with Python but encountered difficulties due to the lack of concurrency support, additional runtime errors resulting from the absence of a compilation stage, and the absence of real-time guarantees. Due to these shortcomings, the research group implemented their solution in Elixir, utilizing Nerves. They saw the benefits of concurrency support at the language level, with structures providing fault tolerance, hot-code loading, and efficient syntax. They also found BEAM’s soft real-time guarantees to be suitable for their small power-plant-related use cases. The SCADA-level components of the developed application enabled the integration of multiple local systems at a higher level, forming a distributed system.

Kopentenski and Van Roy [19], in their research, built a decentralized system based on GRiSP nodes. Their aim was to create a setup without any nodes assuming a central role. Their endpoints were capable of setting up network connections and forming clusters with each other on their own, automatically. Their system was also fault-tolerant, capable of handling the uncertain availability of other nodes. They succeeded in building a distributed system that included fault tolerance and distributed storage.

Kalbusch, Verpoten, and Van Roy, in their research [20], also investigated GRiSP’s use. Their research focused on building Hera and using a sensor fusion framework developed for Erlang. In their work, they highlighted the system’s fault tolerance for issues on different scales. Crashes of measurement, data persistence, and data coordination processes did not affect the stability of one another, and the system remained functional upon the recovery of any failed processes. The system consisted of several boards, and as a whole, it could recover even from the complete failure of any of the boards. They were particularly interested in the computational capabilities of GRiSP and Erlang on the edge, as they sought to avoid any cloud connectivity. Cloud connectivity introduces additional dependencies and instability to a system, as a cloud-facing connection may not always function at an acceptable level. In our work, we also find that being able to perform computations locally is of added value, and we are interested in the limits Erlang-based edge computation offers.

Branch and Weinstock [21] were researching the use of Elixir through AtomVM as IoT gateways. Their gateways bridged end nodes reachable via LoRa, a low-power wireless communications technology, with MQTT brokers accessible over the Internet. They compared the performance of an Elixir-based implementation to one in C++, and used Raspberry Pi boards and ESP32 microcontroller hardwares for hosting the gateways. The authors found functional programming and Elixir, particularly, a convenient fit for embedded development, but highlighted the scarcity of libraries that support development for microcontrollers. When comparing the performance of the C++ and Elixir gateways they developed, they identified data loss issues in the Elixir implementation. As in our work, we also investigate the cost of transmission with LoRa. We will identify the configurations that provide stable connectivity and those at which transmission starts to become unreliable.

Stritzinger [22] built a stack for the OPC-UA protocol in Erlang. OPC-UA is a standard used in industrial automation for data exchange between devices. The standard defines the services and protocols for communication among machines on the field and between field devices, business networks or the cloud. It also includes a model for defining data, allowing it to be interpreted by all components that follow the standard.

BEAM’s scalability performance was investigated and improved upon in research by Tinder, Chechina and their colleagues [23]. Their work summarizes a wide effort aiming to understand Erlang’s scalability limits in both single-node and distributed scenarios through experiments. After identifying the bottlenecks, they also improved the scalability of the runtime and developed a set of libraries to support the implementation of distributed systems.

As the study includes single-node benchmarks, they may serve as a basis for further analysis and testing of AtomVM’s performance. Their work analyses the performance on powerful multi-core computers hosting several process schedulers. We found that adjusting their research methodology to analyze AtomVM would be an interesting direction for continuing our research.

Li, Trinder, and their colleagues [24] investigated the incorporation of fault tolerance into IoT systems. In their work, they used the Elixir language throughout the complete system as a tierless language. This allowed engineers to add fault tolerance to the system using BEAM’s native features. Although using BEAM languages comes at the cost of a higher memory footprint, they also found that using higher-level programming constructs and fewer languages for development allowed engineers to work more efficiently.

Our work complements their research by using Erlang and by analyzing the resource footprint and scalability of the language on microcontrollers.

Pereira, Couto and their research team measured programming language efficiency in a thoroughgoing study [25]. They investigated the execution time, energy consumption and memory footprint of different languages solving the same set of problems on a desktop-grade computer. They found that automatically choosing the best language, considering all three measured attributes, is not possible, and the developer should take into account the nature of the developed program when choosing a language. The study ranked Erlang lower than C and C++ in terms of execution time, energy consumption, and memory footprint. However, the study does not focus on the distinctive features of the analyzed languages, and how they fit non-functional requirements. In Erlang’s case, such features are its scalability and the ease of building fault-tolerant systems.

Our work focuses on measuring the footprint of Erlang’s distinctive constructs on an embedded system rather than comparing its performance to that of other languages. However, we think that such a comparison, with a special focus on embedded systems, would be valuable for future work. Investigating the execution times of Erlang algorithms running on microcontrollers would also be an interesting area of research.

## 3. Materials and Methods

In this section, we outline the equipment used for our measurements and the objectives of our tests. We also present the software we developed for driving our microcontrollers, controlling our measurements, and our experiences using AtomVM. Our software is open-source and is publicly available online (https://gitlab.com/d-ferenczi/atomvm-measurements, (accessed on 30 July 2025)).

### 3.1. Test Equipment

AtomVM is compatible with various microcontrollers: devices from the ESP32, STM32 and Raspberry Pi Pico families. Among these options, we have chosen the ESP32 chip for our experiments because it supports the broadest range of features on the platform. These features include APIs for deep sleep, an analogue-to-digital converter, system information for runtime monitoring of memory use, networking, and support for UART (Universal Asynchronous Receiver/Transmitter) serial communication, which we used for interacting with our measurement program. It also supports LoRa (Long Range) connectivity, a radio communication method, typically used to transmit low amounts of data over typical distances of up to 10 kilometers.

The broader range of supported features makes the ESP32 a good candidate for future experiments involving the development of distributed systems and building prototypes for real-world systems. We have gathered the features AtomVM supports on embedded platforms in Table 2, highlighting those required for our experiments.

LoRa, a wireless communications technology, has a low power draw and is often used with devices that consume low amounts of energy. Power efficiency makes the technique popular with battery-powered devices [26]. The technology operates in the unlicensed radio spectrum. It uses “chirps”, linear changes in frequency over time, for encoding the data [27]. Under ideal conditions and without obstacles, LoRa can cover ranges of multiple kilometers. This makes the technology particularly appealing for covering rural areas. As obstacles in the line of sight limit LoRa’s covered range, in urban deployments, the technology is used to cover a shorter range, such as in the case of smart buildings.

LoRa allows developers to customize range, data rate and, consequently, the time on air in their application’s source code, even during runtime. This allows configuring systems optimally for particular deployments.

The ESP32 family of microcontrollers (https://docs.espressif.com/projects/esp-idf/en/latest/esp32/index.html, (accessed on 30 July 2025)) has been developed by Espressif. Development boards come with different feature sets, ranging from the number of cores and instruction set architecture to instructions for machine learning computations, Bluetooth, WiFi, and other connectivity options, as well as the amount of memory. For our work, we have chosen the “WiFi LoRa 32(V3)” (https://docs.heltec.org/en/node/esp32/wifi_lora_32/index.html, (accessed on 30 July 2025)) and “Wireless Tracker” (https://docs.heltec.org/en/node/esp32/wireless_tracker/index.html, (accessed on 30 July 2025)) development boards of Heltec Automation, both of which use the ESP32-S3 variant of the microcontroller family. This model features two physical cores and 512 kilobytes of RAM for applications to use.

We have chosen this particular device because it features WiFi and rich serial connectivity options, which are necessary for interacting with the running application. It also supports LoRa communication, which we have experimented with and plan to utilize for future prototyping as well. The “Wireless Tracker” also features a GNSS chipset, which we plan to use to investigate location-related use cases.

As noted in Trinder’s research [13], which compares C++ and Erlang implementations of telecommunications software, a drawback of the Erlang implementation was its higher memory footprint. As microcontrollers have a limited amount of on-board memory, being able to report the amount of memory consumed by various activities is important. We therefore used the AtomVM’s ESP32 API to report the consumed RAM periodically, and after specific events of interest.

To monitor power draw, we used the TC66 tester (https://fccid.io/2A5Y7-TC66C/User-Manual/User-Manual-5806949.pdf, (accessed on 30 July 2025)) from Hangzhou Ruideng Technologies Co. The device can be used by connecting it between a computer and a board being analyzed. This allows us to establish a UART connection to our device through USB to send arguments for our tests and to receive any data, for example, the reported memory footprint or the number of running processes. The tester device also reports power measurements through a USB connection, which can be read and processed using a script. The device reports current draw every 100 ms, with a measurement resolution of 0.01 mA. We chose this device based on its reporting capabilities, ease of use, and built-in display, which allows for easy reading of the actual consumption.

We planned a convenient measurement workflow that allows for an easy parametrization of our experiments and reports current draw, memory statistics and console output continuously on our PC. This also allows us to see the effects of starting additional processes or transmitting LoRa messages in real time.

Figure 1 presents this test setup: the analyzed device is powered through the tester, and the same USB connection is used to transfer data and flash the software we are using to test a given functionality. We designed our measurement software to be parametrizable through the UART connection, allowing us to configure it from our computer’s keyboard without needing to restart or re-flash the system under test. We found that the UART connection has a consistent, low current draw without a noticeable impact on our power measurements.

We also experimented with connecting two AtomVM nodes via LoRa. For this setup, we wrote software for the sender and receiver roles. By connecting our computer to both devices simultaneously, we were able to monitor the events of both devices. Figure 2 presents this setup, while Figure 3 show our development environments in use.

We also built an implementation of our LoRa sender in C++, a language more commonly used to build efficient software for embedded systems. We use this variant as a reference, to which we compare our Erlang-based implementation. The source code for this baseline implementation is based on the LoRa examples published (https://github.com/HelTecAutomation/Heltec_ESP32/tree/master, (accessed on 30 July 2025)) by the vendor of our embedded boards.

### 3.2. Measurement Goals and Experiments

The high-level goal of our tests is to evaluate the resource use of the features that make Erlang distinctive. Two important building blocks of Erlang’s fault tolerance are lightweight processes and the monitor or link relationship that can be established between them.

#### 3.2.1. Process Scalability and Management

Erlang processes are lightweight, as they have a short memory footprint, making them more scalable than operating system threads. They are handled by the Erlang runtime and have garbage collection processes that are specific to them. An Erlang system is capable of starting several thousand processes in a fraction of a second.

Erlang processes can be set in various supervision relationships with one another. This allows processes to take a supervision role to relaunch crashed processes at a moment’s notice. The exact steps for failure handling depend, of course, on each scenario and require thorough planning from the developer. Erlang offers various mechanisms for establishing connections between processes. ‘Monitors’ provide messages to supervision processes whenever a monitored process terminates. ‘Links’ declare a tighter coupling between two processes: by default, a crash of a linked process will crash the other process as well. Finally, the ‘supervisor’ is a design pattern offered by Erlang’s standard library. Supervisors allow easy configuration of monitoring behavior, e.g., whether a supervisor should restart all supervised processes if one of them crashes, or the number of times it should try restarting processes before terminating itself.

These features, combined, allow for building fault-tolerant applications, as even though faults can lead to an unstable state in a process and eventually cause a crash, they can be restarted promptly without any noticeable impact. To assess the extent to which AtomVM facilitates the creation of fault-tolerant systems on the edge, it is crucial to determine the limitations the platform imposes on utilizing these features. To test this, we designed the following experiments:Measure the number of processes that fit in memory.Measure the footprint of processes with different monitoring types.Measure the footprint of data processing strategies.

In all experiments, we measured the number of processes that could be started together, along with their total power and memory consumption.

We made process initialization dynamic whenever possible: the number of active processes was always set during runtime. This allowed us to conduct experiments without needing to re-flash the device. To ensure the scheduler maintains all processes, we wrote a small piece of code for the firmware to measure the voltage on a specific, unconnected pin of the device at set intervals. We highlight that for our measurement, the values read on the pins are of no relevance; the task’s sole role is to keep the processes busy.

We emphasize the value of parameterizing our experiments during runtime through the serial interface. Although it does not offer the level of convenience of an automated test pipeline, being able to set the number of processes and inspect the memory and power impact simultaneously made experimenting more practical. More elaborate power testers that feature automatic plotting and marking of charts based on GPIO signals sent from the device would also make data processing more convenient.

To replicate how gathered data are processed on actual edge devices, we also examined different aggregation strategies that are well-suited for low-power embedded devices.

#### 3.2.2. Data Transmission

Real setups also require communication between other devices or central components of the system. For this scenario, we based our experiments on the LoRa technology.

The power requirements of LoRa communication primarily depend on the transmission configuration. The power consumption of a LoRa setup depends on several parameters. Transmission power directly affects consumption, while bandwidth and spreading factor determine the time required to send a set amount of data and, consequently, the power needed for transmission. An optimal configuration of LoRa depends on the use case, typically the distance between devices, as well as the available power sources, and has its own field of study.

Our experiments can be configured with the following parameters:Payload size of transmitted message.Time elapsed between transmissions.

Apart from our focus on memory and power footprint, we were also interested in the speed and stability with which AtomVM can work with LoRa. We conducted further experiments with different transfer frequencies, keeping in mind that past research [21] suggested data loss at speeds higher than one message per second when using AtomVM. In our experiments, we aimed to identify thresholds specific to AtomVM, independent of LoRa, which would help determine the practical use of the technology.

We also measured the performance of a C++ implementation under the same configuration. This comparison allows us to better understand the tradeoffs and benefits of using Erlang.

We measured the stability of the connection between our devices by increasing a counter value on the sender end, which we included in the sent package. On the receiver end, we counted the received messages to programmatically observe any difference between the number of packages sent and received. Such a difference would still need to be debugged, of course, as there could be many reasons for it, ranging from signal interference to sending or receiving data more frequently than the underlying drivers can support.

## 4. Results

In this section, we present the results of our measurements for managing and supervising processes, followed by our evaluation on sending data through LoRa. We continue with our observations regarding AtomVM’s fault tolerance and our recommendations for using the runtime.

The raw data of our measurements is available at our source repository (https://gitlab.com/d-ferenczi/atomvm-measurements/-/blob/master/data.tar.gz, (accessed on 30 July 2025)).

### 4.1. Process Scalability and Management

We began our experiments by starting new processes that sample the voltage on a given pin every 50 ms. We set the sampling frequency to a low level to keep processes working continuously on a computationally inexpensive task. We observed that small increases in power consumption occurred as processes were started, first reaching a limit at approximately 64 mA, when we initiated two additional processes. Subsequent processes lowered the total current draw, which stabilized around 58 mA independently of the number of additional processes. Given that the device has two physical cores, we assume that this behavior is due to better resource utilization on concurrency levels close to the physical limit.

In total, we were able to fit around 370 unmonitored processes in memory. Adding monitoring increased the memory footprint of each process. In total, we ran around 350 monitored and 360 linked processes on our devices. We also observed current spikes of 64 mA for monitored processes.

To verify consistent behavior, we only increased process counts after monitoring resource use for one minute to gather multiple measurements. We also ran each experiment 10 times and found that the readings were constant, with no significant outliers.

We implemented the same workers as “server behavior” processes supervised with a “supervisor behavior”, a construct from Erlang’s standard library, OTP. This pattern enables the setting of fine-grained actions for controlled processes, taking into account adequate restart strategies that align with the dependencies between processes. We also observed power draw spiking at a low number of six workers. In total, we could fit fewer servers, about 195 OTP-supervised ones in memory, than simple processes with manually set monitors, even though the remaining 83 kilobytes should have left room for additional server processes. The OTP-supervised workers drew about 1 milliampere less current than those that were not monitored, or supervised manually with monitors or links.

Figure 4 and Figure 5 present our measurement results for remaining memory and current draw, respectively. Table 3 summarizes our measurements.

We saw this difference as surprising, as we expected the more feature-rich OTP behaviors to consume more power. To verify whether there was a difference in performance, we conducted an experiment with workers repeatedly calculating the 22nd Fibonacci number concurrently on 12 and 25 processes. We found that OTP-supervised processes consume approximately 1 milliampere more power and calculate the configured Fibonacci number about 13 ms faster than their unmonitored counterparts. These measurements suggest that the child processes of OTP supervisors are better at leveraging available resources.

It is important to note that OTP servers abstract away the implementation details used for continuously running processes. Manually written, continuously running processes require developers to write explicit, recursive calls, while OTP servers abstract this point. We assume that the differences in our measurements are due to implementation differences between the OTP and manual implementation in waiting for and performing recursive calls.

Of course, in practice, it is unrealistic to run hundreds of worker processes on a two-core microcontroller. However, in Erlang, processes responsible for connections or maintaining state, in general, are crucial components that enable both encapsulation and fault tolerance. Depending on the memory requirements of a given use case, even OTP behaviors can be implemented for added control and fault tolerance.

For comparison, we can examine the number of processes that run on AtomVM in relation to those supported on computers. Erlang’s full-featured distribution allows setting the maximum number of simultaneously running processes to 134,217,727 (https://www.erlang.org/doc/apps/erts/erl_cmd#max_processes, (accessed on 30 July 2025)). A maximum number of 134 million lightweight processes tracking connections is of great value for handling scaling challenges in mission-critical systems. On embedded systems, we do not expect scaling problems of such magnitude. Apart from encapsulating the state of active communication channels, we find their worth in building fault-tolerant process hierarchies that help reduce code volume in the narrow task a microcontroller performs.

During our work, we identified a memory leak (https://github.com/atomvm/AtomVM/issues/1384, (accessed on 30 July 2025)) related to freeing memory of stopped processes, which is planned to be addressed in AtomVM’s upcoming release. OTP behaviors also lacked support for dynamically stopping and adding supervised child processes. We saw that this feature is also planned in the runtime’s next edition.

We also extended our experiment based on OTP supervisors with an aggregator process, to which worker processes can forward their data. According to the actor model, Erlang data are transferred between processes via lightweight messaging. In this setup, our aggregator processes collect data by receiving the messages sent by our workers.

We defined three different strategies for our aggregator:Forwarding each received datapoint through UART.Performing a simple aggregation on the data: finding the minimum, the maximum and the average of the received data.Collecting data in batches and forwarding these once they reach a given size.

For all experiments, we set 10 workers reading voltages from a given pin of the board every 500 ms by default. We found that maintaining the original sampling frequency of 50 ms led to the aggregator receiving more messages than it could process, which ultimately resulted in the system running out of memory and encountering runtime errors. We tried different sampling frequencies in our 10-worker setup and found those above 300 ms sustainable. Developers expecting their applications to handle high-frequency messaging between processes for extended periods can monitor the size of their message queue during development.

For the batching experiment, we tried different batch sizes of 70 and 300 samples but saw no substantial difference between their footprints.

For each strategy, we measured the power draw at least 80 times and averaged the results to obtain a more accurate representation. All four scenarios had a median around 41 mA. Concerning memory, only batch processing had a lower footprint by a few hundred bytes. We also observed greater variance in their memory usage, indicating how data batches are being filled before they are sent.

We monitored resource use for a minute for each experiment and re-executed them five times. We found their behavior consistent, without significant differences between test runs.

Our measurements are summarized in Table 4.

In summary, we found that actual concurrent applications with separate worker and aggregator processes require testing to verify if the speed at which data are sent to the aggregator is adequate. We saw no substantial difference in data processing strategies on their own. Forwarding data from the aggregator was tested with UART, which has a relatively cheap footprint, as seen by comparing the setup without aggregation to the one that simply forwards data. In practice, data may need to be transmitted wirelessly, which requires more energy. Such a condition may require sending data less frequently and selecting an aggregation strategy that meets this requirement. In the next subsection, we present our findings regarding the transmission of data using LoRa.

### 4.2. Data Transmission

To evaluate a LoRa-based setup, one key metric is airtime: the time a single device spends transmitting data. Airtime directly affects battery use, as devices consume more power while actively transmitting. Airtime, apart from depending on the amount of data sent, also depends on the particular LoRa connection’s details. The spreading factor is directly proportional, while the bandwidth is inversely proportional to the airtime. These parameters also affect the signal range and should be adjusted according to the use case.

Airtime is also subject to regulation. In the European Union, access to radio spectrums between 25 Mhz and 1000 Mhz is regulated by the European Telecommunications Standards Institute’s (ETSI) ETSI EN 300 220-2 standard (https://www.etsi.org/deliver/etsi_en/300200_300299/30022002/03.02.01_60/en_30022002v030201p.pdf, (accessed on 30 July 2025)). Section 4.3.3 of the standard specifies the length of idle intervals that have to pass between transmissions. The proportion between airtime and idle intervals is referred to as duty cycles. Duty cycles also vary depending on the specific frequency sub-band used and range between 0.1 % and 10 %. In practice, this means that, in the case of a 10 % duty cycle, if a transmission of 16 bytes requires 70 ms airtime, the device will have to stop transmission for about 7 s before sending the next frame. Calculators (https://loratools.nl/#/airtime, (accessed on 30 July 2025)) assist developers in calculating airtime and duty cycles for different setups.

Branch and Weinstock’s experiments [21] measured high data loss volumes at frequencies around 333 ms. We configured our environment to work with a spreading factor of 8 and a bandwidth of 250 kHz. We also sent messages of different sizes, ranging from 1 byte to 64 bytes, and found our transmission to be stable.

To measure the speed at which AtomVM’s LoRa stack is capable of sending and processing messages, we sought to determine the lowest sending interval at which data are consistently received. We continued using our setup with a spreading factor of 8 and a bandwidth of 250 kHz, sending single bytes. We found our transmission stable when sending data every 290 ms. At this point, we also experimented with different payload sizes and encountered data loss when sending more than 27 bytes at a time. Setting the sending frequency to 350 ms resolved this issue, suggesting a performance bottleneck in the AtomVM LoRa driver.

In setups without data loss, we had our measurements running for at least 20 min, after which we compared the number of sent and processed messages.

We also performed these measurements with a sender implemented in C++. Similarly to the AtomVM-based stack, we observed data loss when sending single bytes at a rate of 70 ms. This hints that data are lost on the receiver end of the system. Using a radio-frequency detector would allow us to verify this without the overhead that the LoRa driver puts on the system.

However, we did not observe any data loss when sending payloads of 27 bytes with our C++ sender. As we increased the payload size to 100 bytes, we observed that although the actual transmission became slower than the 290 ms we had configured, the sent data still arrived at the destination. While the limits of transmission are determined by the capabilities of the hardware, the Erlang and C++ LoRa drivers are different in this regard. While the former appears to discard data prepared for sending if a transmission is already in progress, the C++ implementation blocks the code’s sequential execution until the transmission is finished.

Our results are summarized in Table 5.

We find that, in general, the limits are set by the AtomVM stack’s speed to handle LoRa messaging. There may be room for improvement in the implementation. However, we should note that in practice, even sending a 24-byte message through such a configuration requires approximately 51 ms of airtime and has a duty cycle of five seconds.

In real applications, a single server receives data from multiple nodes. A large fleet of devices sending data every couple of seconds will require a server capable of receiving these frames at a frequency proportional to the number of nodes. Configuring the appropriate intervals depends on each use case, but AtomVM’s LoRa driver helps with this effort by providing constructs that facilitate synchronous, fault-tolerant communication between nodes.

LoRa’s practical applications, as researched by Augustin, Yi, and their colleagues [28], however, suggest that the technology is not used for sending data frequently, and communication errors occur rather as a consequence of collisions than due to performance bottlenecks of the senders or receivers.

To investigate the consequences of collisions, we set up two nodes to send data to a single receiver every 300 ms. As a consequence of the small intervals between transmissions, we observed collisions immediately after launching the system. However, Erlang’s fault tolerance helped us handle corrupt data with minimal development effort.

We found that memory use is only dependent on the size of our payload. The additional memory space taken is in alignment with the memory needed for allocating larger variables. Changing the sending frequency had no effect on memory use.

Regarding energy footprint, power requirements primarily depend on the duration of LoRa transmissions. On the Erlang implementation, airtime draws a current of 85 mA, while an unused, but active, LoRa stack requires 50 mA. Reflecting on the data processing strategies from the previous section, we think that minimizing transmissions and performing local processing whenever possible is a design that fits well with LoRa’s characteristics. Erlang’s actor model helps encapsulate the concurrent tasks of processing and transmitting data.

We also measured the memory and energy footprint of the C++ sender. We found that the implementation has a lower memory footprint and identical power draw when LoRa is idle. However, radio transmissions require more power, approximately 93 mA under identical settings. We found this result surprising, as we had expected the C++ implementation to be more efficient, given that it is running closer to the actual hardware. Analyzing the root cause of this difference would be an interesting investigation that could possibly reveal a bug.

We also observed that the AtomVM requires an additional 150 kilobytes of memory compared to C++. This requirement should be considered when developing applications that require storing or processing large amounts of data.

We summarize the differences in footprint of the LoRa and C++ stacks in Table 6.

These observations indicate that Erlang setups perform well within the transmission frequencies allowed by duty cycle regulations. Data loss is possible, but we only encountered it when configuring transmission rates unfit for LoRa applications. In practice, errors stemming from collisions are more common, but these are independent of the language used to implement software on the nodes. Erlang’s features and higher-level abstractions help handle data corruption in a cleaner and more efficient manner.

### 4.3. Fault Tolerance on AtomVM

Fault tolerance is one of the main features of Erlang. To better understand how AtomVM brings this capability to embedded systems, we analyzed the different runtime errors we encountered in detail. To this end, we classified them into three categories:Erlang process failures.Mailbox overflows.Performance problems.

#### 4.3.1. Erlang Process Failures

Erlang monitors and supervisors offer a solution for the first type of errors: monitoring processes can react on the failure of processes, possibly restarting them in a stable state. The implementation of the restarting logic and monitoring structure still has to be correct, as a faulty implementation will just keep restarting processes in an unstable state.

Allowing Erlang processes to fail with such errors is normal in Erlang applications, as their presence should be handled by well-defined supervision structures.

AtomVM includes the features for building these supervision structures. Configuration options do not currently include all features from BEAM’s Erlang supervisors. However, those available allow for structuring processes in a fault-tolerant manner.

#### 4.3.2. Mailbox Overflows

Erlang processes exchange data by sending messages to one another. These messages are delivered to the recipient’s mailbox. By design, the memory footprint of the mailboxes can grow up to the memory available on the system. This poses a problem if a process receives messages faster than it processes them. In this case, the receiving process will eventually consume all system memory, leading to general application failures.

We encountered this error when sending data too frequently to a central supervisor process. As the application became unresponsive, eventually the device’s watchdog restarted AtomVM, along with our application. This problem can be particularly easy to trigger when using a large number of processes sending data frequently to an aggregator. Such a setup is not only problematic when assigning frequently sampling processes to different hardware components, e.g., sensors, but also when designing algorithms for processing data. A parallel algorithm might easily scale up to hundreds of processes sending their results as Erlang messages to a collector, overflowing its mailbox.

Avoiding overflowing mailboxes is the responsibility of the developer and should be addressed through a thoughtfully designed architecture that prevents higher volumes of messages from being sent for prolonged periods.

Methods for preventing and resolving mailbox overflows [29] include the following:Monitoring queue lengths.Rate-limiting sender processes.Handling messages of any structure, to avoid a constantly growing mailbox.Offloading heavier computation to other processes.

Choosing an adequate code structure for handling overflowing mailboxes depends on the software architecture of the application in question and is outside the scope of our present work.

Addressing this problem requires more care when developing AtomVM applications, as mailboxes have less available memory than applications deployed to BEAM.

Relying on the watchdog for restarting the application is not an adequate strategy. A restart will not only lose any intermediate state but the application may be unresponsive for several minutes before the watchdog restarts the system.

Configuring the watchdog in line with the expected system load is also the responsibility of the developer.

#### 4.3.3. Performance Problems

The system can also become unresponsive if its processing load is too high. Similarly to before, in such an event, AtomVM will fail to restart the system’s watchdog, and the device will eventually be restarted completely. We encountered this problem while experimenting with processes that compute Fibonacci sequences. For example, computing the 28th sequence on multiple processes will hinder the timely restart of the watchdog’s timer and trigger a complete reboot.

We classified this problem separately, as choosing a fitting workload for the hardware is a responsibility independent of the implementation language.

Altogether, we find that Erlang’s lightweight processes and supervision capabilities enable the development of fault-tolerant applications for embedded systems. However, the lower amount of available memory also requires more care from developers. Message passing, central to the architecture of any Erlang application, must be designed with particular care to avoid overflowing mailboxes.

### 4.4. Developing on AtomVM

As Erlang developers, we found some difficulties in using AtomVM. Particularly, OTP behaviors are implemented with some of their features missing. This makes porting applications more challenging, as developers must refactor their code to use only functions included in the runtime. Some missing features, like the dynamic supervision of processes with OTP, are already implemented and planned for a future release.

We found the experience of working with Erlang for embedded development swift compared to the more popular C++ and Arudino IDE-based tools (https://docs.arduino.cc/learn/starting-guide/the-arduino-software-ide/, (accessed on 30 July 2025)). Working with a dynamic language that allows developers to set aside defensive programming enables us to work faster. Fast compilation times and the ease of automating CLI-based tooling also allowed us to work more efficiently.

For implemented features, we found the source repositories and documentation especially helpful, as they contain many examples that focus on different Erlang features or hardware components of a developer board. As developers, we found it easy to build on these examples to create our test applications.

We foremost recommend AtomVM for applications that
Perform multiple tasks, and would benefit from encapsulating these as independent processes.Require scalability.Are complex, and can be implemented better using the higher-level constructs of functional programming.Would otherwise require elaborate defensive programming.Benefit from binary pattern matching.

We would recommend using C++ for applications that require a higher amount of memory. We have seen that C++ implementations leave an additional 140 kilobytes of memory for the developer. We also find C++ a better fit for simple, single-threaded applications that do not require scaling and do not benefit from encapsulated processes.

In terms of power draw, we found C++ to be marginally more efficient than AtomVM when not transmitting data. Although we found AtomVM more efficient when using LoRa, we attribute this difference to a software or configuration issue, rather than a difference in inherent efficiency. Therefore, we find C++ a better fit for cases that require an optimized power draw.

## 5. Discussion

Data privacy concerns in IoT have sparked interest in technologies that support local data processing without a connection to the cloud. AtomVM supports this effort by bringing a language typically used for server applications to embedded systems driving sensors. Even though these devices typically offer only one or two computing cores, Erlang constructs make it possible to encapsulate logic intended to run in parallel into separate processes. Processes have a small footprint; a system with only 512 kilobytes of RAM can host several hundred of them, depending on the supervision features used.

These features enable developers to add fault tolerance to their systems. We consider this an important contribution, as embedded development is typically constrained by test environments that are difficult to automate and expensive to deploy bug fixes.

Past research has already investigated Erlang’s use in IoT on more powerful devices. We looked at multiple examples that performed elaborate computations: sensor fusion [20], soft real-time control [18], and mission-critical communication [17]. These research efforts developed libraries that run on full-featured Erlang distributions on high-powered devices.

In our future work, we plan to investigate whether these libraries can run on AtomVM’s restricted feature set, possibly port them, and evaluate their performance.

We also plan to continue our work by developing a prototype for a given use case. We find challenges in precision agriculture and precision livestock farming particularly interesting, as they involve power efficiency, long-range communication, unstable connectivity, and the practicality of local data processing. Apart from privacy concerns, in these use cases, endpoints are often disconnected from high-bandwidth networks, and moving large volumes of data to central components is not feasible. Although technologies like LoRa help transmit over large distances, battery power can be saved by sending data less often. Embedded systems’ and LoRa’s use in precision agriculture has been researched in multiple studies.

McIntosh, Cibils and their colleagues [30] used LoRa and GNSS to monitor the position of grazing livestock. Their system tracked the position of cattle in real time to detect patterns in animal behavior, e.g., the frequency at which they visit a water tank, at which locations they preferred camping or even possible accidents, and when animals were not moving for extended periods. Communication was based on LoRaWAN, a protocol based on LoRa that defines device roles for moving data to a centralized location but disallows direct device-to-device communication. On average, they were able to deliver 46% of GPS data. Although it would require a more elaborate design, we believe that using LoRa to enable transmissions between devices would help increase the delivery rate.

LoRa was also used by dos Reis, Easton, White and Fuka [31]. They also created a bespoke system with an embedded board, an accelerometer and a GPS receiver. Their implementation consisted of averaging measured data collected in 15-s time windows. They measured data loss between 40% and 60%, highlighting that the cause of these high losses may be the high frequency at which data are sent, and that a better design of sampling and transmission frequency would improve data transfer reliability.

Karl and Sprinkle [32] researched the development of trackers to log animal positions using commercial off-the-shelf parts. Along with the low cost of approximately USD 55, they highlight the issues they faced due to faulty soldering, which caused increased power draw on some occasions and hardware failures. We find that Erlang’s fault tolerance can be helpful when prototyping with readily available components.

Regarding the detection of accidents or illnesses that animals may suffer, Papakonstantinou and Voulgarakis [33] and their colleagues surveyed precision agriculture technologies that enhance animal welfare. They highlight the problem of detecting diseases as early as possible. Although health problems can be inferred from location data, we can measure these more accurately by placing designated sensors on different parts of the animal. Different sensors responsible for a given task typically communicate independently with central systems. Aggregating data on the animal itself might help manage the different systems, and sensor fusion would help derive more accurate observations on the edge.

In general, current challenges in precision agriculture [34] relate to increasing the number of tasks performed on the end device or the gateway. Such functions include processing data locally, managing multiple physical sensors and compressing data.

Microcontrollers are also used for concurrent tasks in other domains. Bolbi, Salman and their colleagues [35] built a device for hybrid, LoRa and optical-based communication. Their microcontroller-based system forwards data received through its optical channel via LoRa and is also equipped to communicate in the opposite direction.

Pestana, Radeta and their colleagues [36] prototyped a microcontroller and LoRa-based system for marine monitoring. Their devices retrieved GNSS data, processed it locally to reduce its size, and forwarded it through LoRa to gateway nodes. They implemented an elaborate process for preparing and sending the data in brief time intervals.

These studies outline the requirements for concurrent architectures across various domains. End devices simultaneously gather data from various sources while managing computations and transferring data to other devices. Additionally, we must account for failure on all scales: sensors or even a complete node may malfunction, and communication may not always be reliable between devices. Our results suggest that Erlang’s support for encapsulating tasks in concurrent processes, combined with an actor model, can be utilized for building fault-tolerant structures on the edge. We also found that working on our measurement software was faster in Erlang than in C++. Erlang’s faster compilation times, dynamic types, and ability to focus on our code’s ’happy path’ also allowed us to work more efficiently. Even considering the missing features of AtomVM, Erlang brings benefits to developing embedded systems, which are worth exploring in a prototype.

## 6. Conclusions

In our work, we examined how Erlang can address common challenges in developing embedded systems. Erlang is a language designed for building fault-tolerant distributed systems, typically used for implementing telecommunications, banking and other mission-critical systems. We consider distribution and demand for fault tolerance a common denominator between these industries and IoT setups.

AtomVM is a reimplementation of the Erlang runtime designed for use on embedded boards. Although it contains the main features for building resilient distributed systems, there are missing and partially implemented features, and those available are constrained by the limited power of microcontrollers.

To investigate whether AtomVM is a suitable candidate for building actual prototypes, we conducted measurements to evaluate how Erlang’s concurrency and process monitoring capabilities perform on it. We also assessed its stack for using LoRa, a wireless communication method used for bridging long distances.

We measured that microcontrollers can host several hundred lightweight Erlang processes, depending on their level of supervision. Altogether, we found the limits for managing processes and processing LoRa communication within the bounds for building fault-tolerant, purpose-made software for embedded devices.

We also compared an AtomVM and a C++ implementation of an application transmitting LoRa messages. We found both variants suitable for processing messages while honoring regulations on LoRa transmission frequency. We also found that the C++ implementation requires less memory and is, therefore, more suitable for applications that process larger amounts of data.

Based on the limits we measured in our current contributions, in our future research, we plan to identify a use case for creating an Erlang-based edge device prototype. We also hope that our work will help industry experts in IoT systems assess whether AtomVM fits the use cases they are researching.

## Figures and Tables

**Figure 1 sensors-25-04843-f001:**
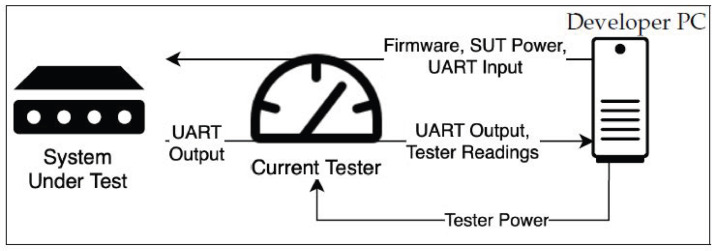
Setup for testing concurrency limits.

**Figure 2 sensors-25-04843-f002:**
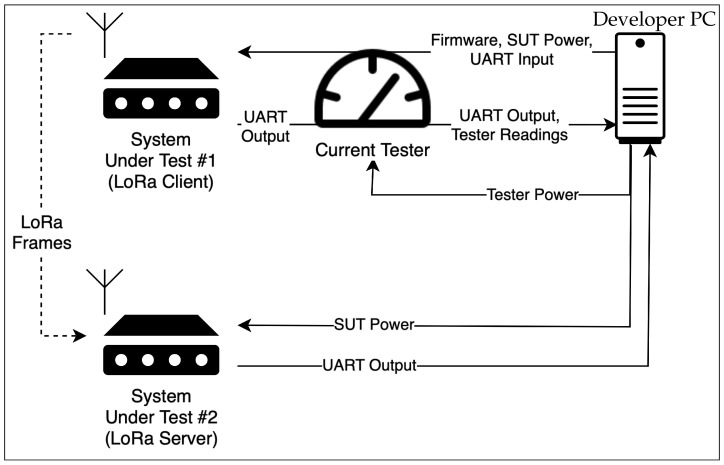
Setup for testing LoRa communication between nodes. System Under Test #1 sends LoRa frames to System Under Test #2, while logs of both devices are captured by the Developer PC.

**Figure 3 sensors-25-04843-f003:**
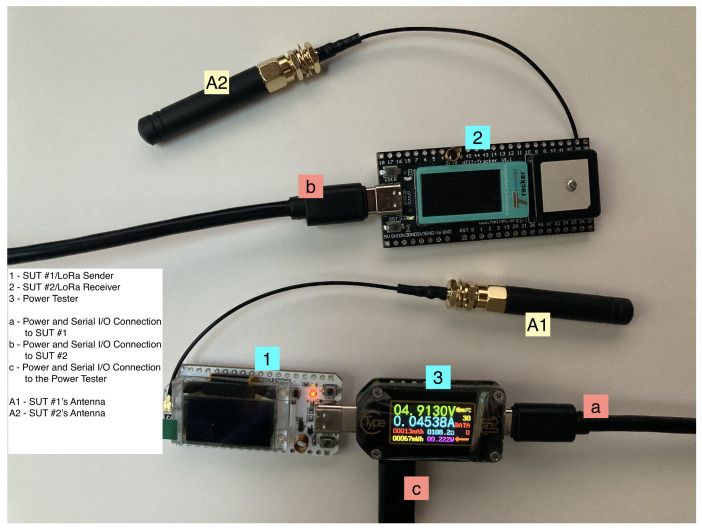
Our development setup consisting of a LoRa sender, a LoRa receiver and a power tester.

**Figure 4 sensors-25-04843-f004:**
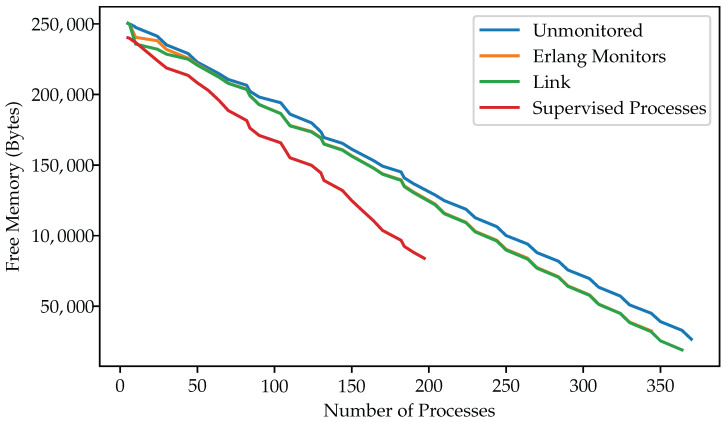
Memory Consumption of Processes under Different Monitoring Methods. Shorter lines reflect how different supervision methods reduce the number of processes that can fit in memory.

**Figure 5 sensors-25-04843-f005:**
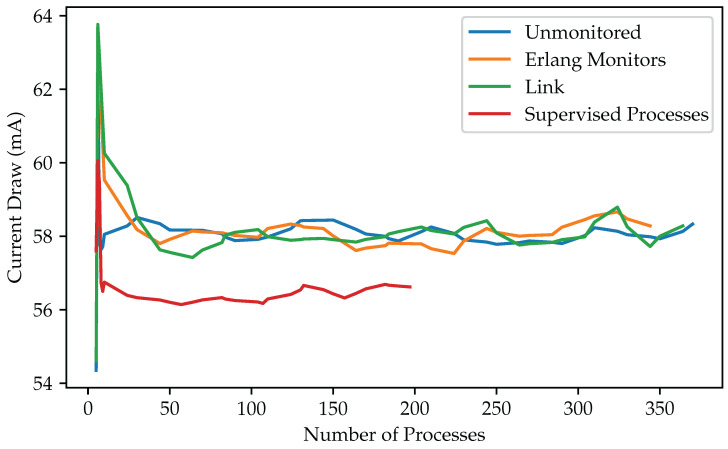
Current Draw of Processes under Different Monitoring Methods. Shorter lines reflect how different supervision methods reduce the number of processes that can fit in memory.

**Table 1 sensors-25-04843-t001:** A summary of how Erlang’s features help build embedded systems.

Challenges of Embedded Systems in IoT	Erlang’s Features
Difficulty of deploying software patches	Avoiding bugs with fault tolerance [15]
Complexity of testing distributed systems	Distributed Erlang, built-in features of Erlang and its standard library
Slow, inefficient development workflow	Dynamic types, fast compilation, smaller code-base [13]
Need for low energy consumption	Erlang distributions designer for low power use ^1^

^1^ According to our measurements, an idle AtomVM instance running on an ESP32 device consumes 39.4 mA. In contrast, an idle C++ application consumes 45.16 mA. The recommended voltage for powering an ESP32 is 3.3 V.

**Table 2 sensors-25-04843-t002:** A summary of the features AtomVM supports on different platforms. List based on AtomVM’s list of supported features ^1^.

Feature	ESP32	STM32	Raspberry Pi Pico
GPIO	Supported	Supported	Supported
ADC	Supported	-	-
I2C	Supported	-	-
SPI	Supported	-	-
UART	Supported	Supported	-
Networking	Supported	-	Supported
LoRa	Supported	-	-
Memory Monitoring API	Supported	-	-

^1^ https://doc.atomvm.org/main/programmers-guide.html, (accessed on 30 July 2025).

**Table 3 sensors-25-04843-t003:** Total amount of processes fitting in memory and their consumption.

Supervision Method	Number of Processes	Average Power Draw (mA)
No supervision	370	58.04
Erlang process monitors	350	58.09
Process linking	360	58
OTP supervision	195	56.48

**Table 4 sensors-25-04843-t004:** Median power draw and memory use of different data processing strategies. All measurements were conducted multiple times, after starting the worker processes, in time windows representing regular application behavior.

Data Processing Strategy	Median Power Draw (mA)	Free Memory (Bytes)
No aggregation	41.25	237,104 ^1^
Direct forwarding of data	41.71	237,220 ^1^
Simple computation	41.64	237,016
Batching in lists of size 70	41.40	236,968 ^1^
Batching in lists of size 300	41.55	236,784 ^1^

^1^ Our measurements showed a few outliers of a few hundred bytes.

**Table 5 sensors-25-04843-t005:** Overloading AtomVM’s LoRa processing through payload size and messaging frequency. To observe data loss, we let our experiments run for at least 30 min.

Sending Frequency (ms)	Payload Size (Bytes)	Airtime (ms) ^1^	Data Loss
70	1	20.74	Data Loss Observed on Both AtomVM and C++
290	1	20.74	No Data Loss
290	16	41.22	No Data Loss
290	24	51.46	No Data Loss
290	27	51.46	Data Loss Observed on AtomVM
350	27	51.46	No Data Loss

^1^ Airtime calculated with https://loratools.nl/#/airtime, (accessed on 30 July 2025).

**Table 6 sensors-25-04843-t006:** Comparing the memory and power footprints of LoRa transmission based on AtomVM and C++.

Software Stack	LoRa Status	Median Power Draw (mA)	Free Memory (Bytes)
AtomVM	Idle	50	210,092
AtomVM	Transmitting	85	210,092
C++	Idle	47	364,220
C++	Transmitting	94	364,220

## Data Availability

Measurement code and results are available at our source code repository: https://gitlab.com/d-ferenczi/atomvm-measurement/-/blob/master/data.tar.gz, (accessed on 30 July 2025).

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
