# Peer review of "Measuring Erlang-Based Scalability and Fault Tolerance on the Edge"

_sensors, 2025, doi:10.3390/s25154843_

Round 1
Reviewer 1 Report
Comments and Suggestions for Authors
- It is recommended that Table 1 be enhanced by including more specific technical data. For instance, no actual consumption values or power ranges are provided, while low power consumption is mentioned. Including such details would allow for a more transparent and objective assessment.
- A more detailed comparison highlighting the rationale behind selecting the ESP32 over other devices is encouraged. Additionally, a comparative table summarizing the key technical features of each option should be included to strengthen this justification.
- Presenting the measurement data in graphical form would improve the clarity of the experimental results and facilitate a more intuitive understanding of the observed processes.
- Similarly, the current measurements should also be visualized using graphs. Those dates support a more effective and straightforward interpretation of the electrical data.
- It is recommended that the references section be strengthened by incorporating recent publications from high-impact journals, preferably from the last five to six years. These changes would enhance the study's scientific relevance and support.
- A substantial restructuring of the manuscript is advised. Improvements should be made in the writing style, the logical flow of ideas, the development of the state of the art, the review of related work, and especially the articulation of the study's original contributions.
- Additional diagrams related to the measurements and visual evidence of the tests conducted are also encouraged. This would contribute to a more complete and robust presentation of the experimental work.
Author Response
Comment 1: It is recommended that Table 1 be enhanced by including more specific technical data. For instance, no actual consumption values or power ranges are provided, while low power consumption is mentioned. Including such details would allow for a more transparent and objective assessment.
Response 1: We thank the reviewer for their advice. We have linked related references and made a note on power consumption based on a comparison between AtomVM and C++.
Comments 2: A more detailed comparison highlighting the rationale behind selecting the ESP32 over other devices is encouraged. Additionally, a comparative table summarizing the key technical features of each option should be included to strengthen this justification.
Response 2: Such a comparison is of value, as not even the official documentation contains a comprehensive list of features supported for each platform. We have gathered a list of supported features in Table 2, and expanded our justification for choosing ESP32.
Comments 3: Presenting the measurement data in graphical form would improve the clarity of the experimental results and facilitate a more intuitive understanding of the observed processes.
Response 3: We agree, a plot can show our results more clearly. We have created charts on our gathered data to improve the clarity of our results.
Comments 4: Similarly, the current measurements should also be visualized using graphs. Those dates support a more effective and straightforward interpretation of the electrical data.
Response 4: For a clear presentation, we have plotted our data for current and memory measurements.
Comments 5: It is recommended that the references section be strengthened by incorporating recent publications from high-impact journals, preferably from the last five to six years. These changes would enhance the study's scientific relevance and support.
Response 5: We thank the reviewer for their suggestion. We have included high impact references both for related work and for our discussion. We hope, these references also strengthen our plans for future research.
Comments 6: A substantial restructuring of the manuscript is advised. Improvements should be made in the writing style, the logical flow of ideas, the development of the state of the art, the review of related work, and especially the articulation of the study's original contributions.
Response 6: We thank the reviewer for their observation. We put special emphasis in reviewing and restructuring the introduction, separating the general introduction of our work (Section 1: Introduction), including a clearer statement of our contributions and a more technical presentation of our research's background (Section 2: Background). We hope, these changes improve the clarity of our work.
Comments 7: Additional diagrams related to the measurements and visual evidence of the tests conducted are also encouraged. This would contribute to a more complete and robust presentation of the experimental work.
Response 7: We added an annotated picture of our setup, so that readers can have a clearer understanding of our work environment.
We thank the reviewer for their time and work invested in reading and commenting our paper. We found their feedback valuable for improving our article. For clarity, we also gathered a list of the changes we made in our revision:
- Change title
- Change wording replacing the term 'benchmark'
- Rephrase contributions
- Section 1: Split subsections into separate section
- Section 2: Clarify research background
- Section 2: Add recent references on IoT development practices
- Section 2: Add references on testing BEAM's scalability
- Section 2: Add references on comparing programming languages
- Section 3: Highlight ESP32's advantages for future work
- Section 3.1: Add notes on C++ implementation used as baseline
- Section 3.2.1: Add note on data aggregation strategies
- Section 3.2.1: Add note on test parametrization during runtime and further options for improvement
- Section 3.2.2: Add note on measuring against a C++ implementation
- Section 4: Rephrase section introduction
- Section 4.2: Add comparison with C++ implementation
- Section 4.3: Add subsection on errors
- Section 4.4: Restructure section, emphasizing guidelines
- Section 5: Add recent references on LoRa's applications, highlighting research directions
- Table 1: Add references and subscript note on base power draw of AtomVM
- Table 2: Add summary of AtomVM features available on different embedded platforms
- Table 3: Adjust decimal comma, remove unneeded superscript
- Table 4: Add missing unit to header
- Order abbreviations alphabetically
Reviewer 2 Report
Comments and Suggestions for Authors
The contribution demonstrates how Erlang can address the prevalent challenges encountered in the development of embedded systems. Erlang is a programming language specifically designed for constructing fault-tolerant distributed systems, commonly utilized in telecommunications, banking, and other mission-critical applications. The authors emphasize the importance of distribution and the necessity for fault tolerance as a common denominator across these industries and Internet of Things (IoT) configurations. The subject of benchmarking fault-tolerance concurrency on the edge is important, and the analysis and proposal presented in the paper serve as a foundation for developing solutions and discussions on this topic.
• What is the main question addressed by the research?
The paper, Benchmarking fault-tolerant concurrency on the edge, addresses the primary question of how Erlang can effectively tackle the challenges of developing fault-tolerant and distributed embedded systems. The authors highlight the significance of distribution and fault tolerance, emphasizing the need for benchmarking fault-tolerance concurrency specifically at the edge of networks. This focus aims to establish a foundation for developing solutions and fostering discussions around these critical topics in embedded systems development.
• Do you consider the topic original or relevant to the field? Does it address a specific gap in the field? Please also explain why this is/ is notthe case.
As I said in the review form, it is relevant and original but not exceptional. In effect, the authors provide interesting results and information (for me, personally, it was a surprise that some results used Erlang). The focus on benchmarking fault-tolerance concurrency at the edge is a good contribution.
• What does it add to the subject area compared with other published
material?
This paper aims to establish a foundation for subsequent discussions and solutions within this field. This implies that the authors are tackling an existing gap by not only identifying the issue but also proposing preliminary measures aimed at its resolution, which exemplifies a significant aspect of addressing a research gap.
• What specific improvements should the authors consider regarding the
methodology?
The authors could include a wider variety of real-world scenarios and edge use cases. Drawing from continuous improvement methodologies (as highlighted in the source), the authors could implement iterative testing and validation of their benchmarks. However, as I said, this would allow them to refine their approach based on feedback and ensure their results are reliable and reproducible.
• Are the conclusions consistent with the evidence and arguments presented
and do they address the main question posed? Please also explain why this
is/is not the case.
The conclusions synthesize the key findings. By presenting evidence of Erlang's effectiveness in distributed and edge computing scenarios, the authors ensure their conclusions are grounded in the data and arguments presented in the respective sections of the paper.
• Are the references appropriate?
yes
• Any additional comments on the tables and figures.
In my evaluation, I propose that it can be improved by presenting the results visually (adding to the tables) and by comparing the proposal's achievements.
Finally, as I mentioned in the review, it's not exceptional work, but it contributes interesting elements to the field. It could be published (it's more than a borderline paper) and addresses a very interesting topic within the field.
Author Response
Comment 1: The contribution demonstrates how Erlang can address the prevalent challenges encountered in the development of embedded systems. Erlang is a programming language specifically designed for constructing fault-tolerant distributed systems, commonly utilized in telecommunications, banking, and other mission-critical applications. The authors emphasize the importance of distribution and the necessity for fault tolerance as a common denominator across these industries and Internet of Things (IoT) configurations. The subject of benchmarking fault-tolerance concurrency on the edge is important, and the analysis and proposal presented in the paper serve as a foundation for developing solutions and discussions on this topic.
The paper, Benchmarking fault-tolerant concurrency on the edge, addresses the primary question of how Erlang can effectively tackle the challenges of developing fault-tolerant and distributed embedded systems. The authors highlight the significance of distribution and fault tolerance, emphasizing the need for benchmarking fault-tolerance concurrency specifically at the edge of networks. This focus aims to establish a foundation for developing solutions and fostering discussions around these critical topics in embedded systems development.
Response 1: This is an accurate summary of our paper.
Comment 2: As I said in the review form, it is relevant and original but not exceptional. In effect, the authors provide interesting results and information (for me, personally, it was a surprise that some results used Erlang). The focus on benchmarking fault-tolerance concurrency at the edge is a good contribution.
Response 2: We also consider our work as a step towards further research and prototyping on using Erlang for creating fault-tolerant systems on the edge.
Comment 3: This paper aims to establish a foundation for subsequent discussions and solutions within this field. This implies that the authors are tackling an existing gap by not only identifying the issue but also proposing preliminary measures aimed at its resolution, which exemplifies a significant aspect of addressing a research gap.
Response 3: This is an accurate description of our paper's goals.
Comment 4: The authors could include a wider variety of real-world scenarios and edge use cases. Drawing from continuous improvement methodologies (as highlighted in the source), the authors could implement iterative testing and validation of their benchmarks. However, as I said, this would allow them to refine their approach based on feedback and ensure their results are reliable and reproducible.
Response 4: We thank the reviewer for their suggestion. We agree, that iterative testing and benchmarking is an important step for creating an environment where developers can work confidently. Our present work is limited in this aspect, as even though the tests are parametrizable, a developer still has to flash them on the system, and run scripts to gather results. We highlighted the limits of our work better in the paper, and described the first steps required for improving on them. As we also consider developer experience an important topic, we plan to revisit this task in our future work.
Comment 5: The conclusions synthesize the key findings. By presenting evidence of Erlang's effectiveness in distributed and edge computing scenarios, the authors ensure their conclusions are grounded in the data and arguments presented in the respective sections of the paper.
Response 5: We thank the reviewer for their comment.
Comment 6: In my evaluation, I propose that it can be improved by presenting the results visually (adding to the tables) and by comparing the proposal's achievements.
Response 6: We thank the reviewer for their suggestion. To improve the clarity of our work, we have added charts on our measurement's data, and a picture on our experimental setup.
Finally, as I mentioned in the review, it's not exceptional work, but it contributes interesting elements to the field. It could be published (it's more than a borderline paper) and addresses a very interesting topic within the field.
We thank the reviewer for their time and work invested in reading and commenting our paper. We found their suggestions valuable for improving our work. For clarity, we have gathered a list of changes we made in our revision:
- Change title
- Change wording replacing the term 'benchmark'
- Rephrase contributions
- Section 1: Split subsections into separate section
- Section 2: Clarify research background
- Section 2: Add recent references on IoT development practices
- Section 2: Add references on testing BEAM's scalability
- Section 2: Add references on comparing programming languages
- Section 3: Highlight ESP32's advantages for future work
- Section 3.1: Add notes on C++ implementation used as baseline
- Section 3.2.1: Add note on data aggregation strategies
- Section 3.2.1: Add note on test parametrization during runtime and further options for improvement
- Section 3.2.2: Add note on measuring against a C++ implementation
- Section 4: Rephrase section introduction
- Section 4.2: Add comparison with C++ implementation
- Section 4.3: Add subsection on errors
- Section 4.4: Restructure section, emphasizing guidelines
- Section 5: Add recent references on LoRa's applications, highlighting research directions
- Table 1: Add references and subscript note on base power draw of AtomVM
- Table 2: Add summary of AtomVM features available on different embedded platforms
- Table 3: Adjust decimal comma, remove unneeded superscript
- Table 4: Add missing unit to header
- Order abbreviations alphabetically
Reviewer 3 Report
Comments and Suggestions for Authors
The submitted manuscript focuses on IoT systems and edge computing. The Erlang language for building concurrent and fault-tolerant systems is evaluated. The authors position Erlang as a viable option for implementing embedded system software and for prototyping real-world applications.
First of all, the title of the paper should be more precise. Now, it’s quite general and does not combine well with the real results.
The abstract is concise and clearly states the contributions. The figures are readable. All figures and tables are referenced to in the main body of the paper.
Related work should better be placed in a separate section. In the current form, the introduction takes almost one half of the paper. Afterwards, subsections 1.1 to 1.4 would form a separate section, parts of them can already appear in the introduction to make it more comprehensive. The introduction should also be better supported by the existing literature (references).
The authors write “We conclude with Section 4” (line 75). Indeed, the manuscript has five sections and not four.
It is recommended to add a photograph of the experimental setup.
As the authors conclude that Erlang (AtomVM) can help with common challenges in developing embedded systems, the question arises – what are the guidelines for using it? I mean, to help the readers decide whether it is a good choice for their purpose, some general guidelines would be nice.
The cited literature should be extended, especially to include the most recent works.
Author Response
Comment 1: The submitted manuscript focuses on IoT systems and edge computing. The Erlang language for building concurrent and fault-tolerant systems is evaluated. The authors position Erlang as a viable option for implementing embedded system software and for prototyping real-world applications.
Response 1: This is an accurate summary of our paper.
Comment 2: First of all, the title of the paper should be more precise. Now, it’s quite general and does not combine well with the real results.
Response 2: We thank the reviewer for their suggestion. We have changes the title in our revision, which we hope, that reflects our work and results better.
Comment 3: The abstract is concise and clearly states the contributions. The figures are readable. All figures and tables are referenced to in the main body of the paper.
Response 3: We thank the reviewer for their comment. We have added some changes, listed below, that we hope improve clarity even further for readers.
Comment 4: Related work should better be placed in a separate section. In the current form, the introduction takes almost one half of the paper. Afterwards, subsections 1.1 to 1.4 would form a separate section, parts of them can already appear in the introduction to make it more comprehensive. The introduction should also be better supported by the existing literature (references).
Response 4: We thank the reviewer for highlighting a problem with our paper's introduction, especially with regards to its clarity and lacking support of existing literature. Indeed, the mentioned subsections serve different purposes, and separating them would make our paper's introductory parts clearer.
We have restructured our paper by keeping a shorter, general presentation of our ideas in the introduction (Section 1), while a background section (Section 2) focuses on technical aspects, including reviewed literature. We have also looked for related work to support both sections. We hope that both the introduction and the background are now clearer and better supported for our readers.
Comment 5: The authors write “We conclude with Section 4” (line 75). Indeed, the manuscript has five sections and not four.
Response 5: We thank the reviewer for highlighting this mistake.
Comment 6: It is recommended to add a photograph of the experimental setup.
Response 6: Indeed, a photo helps readers understand better how experiments are performed.
Comment 7: As the authors conclude that Erlang (AtomVM) can help with common challenges in developing embedded systems, the question arises – what are the guidelines for using it? I mean, to help the readers decide whether it is a good choice for their purpose, some general guidelines would be nice.
Response 7: We thank the reviewer for their suggestion. We have listed guidelines based on our results, to support developers in choosing a technology. We have also performed a limited comparison with C++, to highlight some differences between the two languages in this summary.
Comment 8: The cited literature should be extended, especially to include the most recent works.
Response 8: We have reviewed additional papers, related to both our present work, and our future plans. We thank the reviewer for their valuable suggestion, as readers should be able to place our work better among the already existing literature.
We thank the reviewer for their time and work invested in reading and commenting our paper. We found their recommendations valuable for improving our work. For a clear overview of changes in our revision we list our changes below.
- Change title
- Change wording replacing the term 'benchmark'
- Rephrase contributions
- Section 1: Split subsections into separate section
- Section 2: Clarify research background
- Section 2: Add recent references on IoT development practices
- Section 2: Add references on testing BEAM's scalability
- Section 2: Add references on comparing programming languages
- Section 3: Highlight ESP32's advantages for future work
- Section 3.1: Add notes on C++ implementation used as baseline
- Section 3.2.1: Add note on data aggregation strategies
- Section 3.2.1: Add note on test parametrization during runtime and further options for improvement
- Section 3.2.2: Add note on measuring against a C++ implementation
- Section 4: Rephrase section introduction
- Section 4.2: Add comparison with C++ implementation
- Section 4.3: Add subsection on errors
- Section 4.4: Restructure section, emphasizing guidelines
- Section 5: Add recent references on LoRa's applications, highlighting research directions
- Table 1: Add references and subscript note on base power draw of AtomVM
- Table 2: Add summary of AtomVM features available on different embedded platforms
- Table 3: Adjust decimal comma, remove unneeded superscript
- Table 4: Add missing unit to header
- Order abbreviations alphabetically
Reviewer 4 Report
Comments and Suggestions for Authors
The manuscript presents an evaluation of the AtomVM system with an Erlang application for fault-tolerant embedded systems. The paper tests two scenarios: UART test and LoRa test, evaluating their power consumption. A deep bibliographical search is presented along with its discussion.
The main drawback of the presented article is that the limited test scenarios are not sufficient to call it a benchmark; typically, a benchmark evaluates tens of parameters and makes comparisons against a reference system. The presented work only measures two conditions in a very limited environment, which is not enough to draw conclusions about the tested system.
To demonstrate its feasibility, the system must be tested under failure conditions to examine its recovery. The performed tests were too basic and did not represent a real application.
It is recommended to increase the test scenarios to include more metrics and make a comparison against a reference system.
Author Response
Comment 1: The manuscript presents an evaluation of the AtomVM system with an Erlang application for fault-tolerant embedded systems. The paper tests two scenarios: UART test and LoRa test, evaluating their power consumption. A deep bibliographical search is presented along with its discussion.
Response 1: This is an accurate overview of our paper.
Comment 2: The main drawback of the presented article is that the limited test scenarios are not sufficient to call it a benchmark; typically, a benchmark evaluates tens of parameters and makes comparisons against a reference system. The presented work only measures two conditions in a very limited environment, which is not enough to draw conclusions about the tested system.
Response 2: We thank the reviewer for their correction. We have changed the wording in the paper's title and contents to better reflect the evaluation we have performed.
Comment 3: To demonstrate its feasibility, the system must be tested under failure conditions to examine its recovery. The performed tests were too basic and did not represent a real application.
Response 3: To clarify how Erlang and AtomVM can help with managing the faults of an edge device, we have restructured our Results section to present our experience with regards to fault-tolerance separately in an organized manner.
We agree, that AtomVM's handling of real hardware and software fault should also be investigated. Our present work focuses on evaluating those features of Erlang that are building blocks for creating fault-tolerant structures. In our future research we plan to investigate how these structures perform when handling actual faults.
Comment 4: It is recommended to increase the test scenarios to include more metrics and make a comparison against a reference system.
Response 4: We thank the reviewer for their suggestion. As LoRa messaging is a generic subject, we have created a simple implementation in C++ to which we compared the Erlang system. We have also extended our results with the data we obtained from this comparison. We also extended our reviewed literature with a work on comparing the performance of programming languages and a note, on possible future research in this direction. We hope that our comparison will help readers in understanding the benefits and limitations of AtomVM.
We thank the reviewer for investing time in reading and commenting our paper. We found their suggestions valuable for improving our work. For clarity, we have collected the changes made for our revision:
- Change title
- Change wording replacing the term 'benchmark'
- Rephrase contributions
- Section 1: Split subsections into separate section
- Section 2: Clarify research background
- Section 2: Add recent references on IoT development practices
- Section 2: Add references on testing BEAM's scalability
- Section 2: Add references on comparing programming languages
- Section 3: Highlight ESP32's advantages for future work
- Section 3.1: Add notes on C++ implementation used as baseline
- Section 3.2.1: Add note on data aggregation strategies
- Section 3.2.1: Add note on test parametrization during runtime and further options for improvement
- Section 3.2.2: Add note on measuring against a C++ implementation
- Section 4: Rephrase section introduction
- Section 4.2: Add comparison with C++ implementation
- Section 4.3: Add subsection on errors
- Section 4.4: Restructure section, emphasizing guidelines
- Section 5: Add recent references on LoRa's applications, highlighting research directions
- Table 1: Add references and subscript note on base power draw of AtomVM
- Table 2: Add summary of AtomVM features available on different embedded platforms
- Table 3: Adjust decimal comma, remove unneeded superscript
- Table 4: Add missing unit to header
- Order abbreviations alphabetically
Round 2
Reviewer 1 Report
Comments and Suggestions for Authors
The quality of the work presented has improved significantly; however, it is convenient to pay attention to the following points:
1.- It mentions that it is a lower power device than the Raspberry Pi, and that the objective is to use it in low power microcontrollers, which microcontrollers are mentioned, and how much power do they consume compared to the Raspberry Pi?
In Table 1, the current consumption data has already been added; however, it would be essential to add the voltage data or just put the power consumption of each of the devices.
3.- In figures 4 and 5, there is an incomplete measurement in the graph. Place the measurements up to where the complete measurements of all the variables are shown.
4.- Each paragraph should begin with capital letters, and a period should be placed at the end of the paragraph. See page 15 of the document.
5. Although the overflow of the mailboxes is not within the scope of the work, it is advisable to include a paragraph proposing a solution. Otherwise, make it clear that it is not within the scope of the work.
6.- Finally, it is recommended to add an electronic diagram or image of the embedded system that is being proposed.
7.- It should be clarified that LoRa technology is wireless for short distances and without obstacles.
8.- More references were added; however, not all of them are from high-impact journals (JCR) or within the first three quartiles. From the new references, it is recommended to eliminate those that are not from high-impact journals and replace them with others.
9.- In the conclusions, more emphasis should be placed on the value of their contributions.
Author Response
Comments 1: It mentions that it is a lower power device than the Raspberry Pi, and that the objective is to use it in low power microcontrollers, which microcontrollers are mentioned, and how much power do they consume compared to the Raspberry Pi?
Response 1: We thank the reviewer for their suggestion. We added a note on Raspberry Pi 3B's power consumption, and presented more details on how microcontroller power consumption relates to this. We also made a remark, on how sleep functions on microcontrollers help reduce power consumption.
Comments 2: In Table 1, the current consumption data has already been added; however, it would be essential to add the voltage data or just put the power consumption of each of the devices.
Response 2: We expanded the footnote with data on the recommended voltage for powering the embedded device.
Comments 3: In figures 4 and 5, there is an incomplete measurement in the graph. Place the measurements up to where the complete measurements of all the variables are shown.
Response 3: We rearranged the figures for a more comprehensive layout.
We also thank the reviewer for their remark on the incomplete measurements. The original figures were confusing, as they gave the impression of missing data. In practice, Erlang Monitors, Links, and in particular Supervised Processes require more resources. Consequently only a smaller number of these processes can be started. We put more emphasis on this in our Results section. We also corrected the Y axis label on Figure 4.
Comments 4: Each paragraph should begin with capital letters, and a period should be placed at the end of the paragraph. See page 15 of the document.
Response 4: We corrected our paragraphs as per the Reviewer's remark.
Comments 5: Although the overflow of the mailboxes is not within the scope of the work, it is advisable to include a paragraph proposing a solution. Otherwise, make it clear that it is not within the scope of the work.
Response 5: We added a brief list of general solutions, citing a recent reference on details of Erlang's BEAM runtime. We also clarified that the topic is not within the scope of our work.
Comments 6: Finally, it is recommended to add an electronic diagram or image of the embedded system that is being proposed.
Response 6: We added a note to Section 3.1 on that Figures 1, 2, and 3 present diagrams of the system we set up for our measurements.
In the event, that a diagram of another system would help improve our paper, we kindly ask the Reviewer for a clarification about the system that should be represented.
Comments 7: It should be clarified that LoRa technology is wireless for short distances and without obstacles.
Response 7: We added a note on LoRa's characteristics to Section 3.1.
Comments 8: More references were added; however, not all of them are from high-impact journals (JCR) or within the first three quartiles. From the new references, it is recommended to eliminate those that are not from high-impact journals and replace them with others.
Response 8: We searched for further references that support Erlang's benefits, and added a work on the relationship between programming languages and defects from a high-impact journal. We thank the reviewer for their remark, as we found the referenced work a valuable study that relates to our research.
Comments 9: In the conclusions, more emphasis should be placed on the value of their contributions.
Response 9: We revised our conclusion to better reflect our results, and their value in future work.
Reviewer 3 Report
Comments and Suggestions for Authors
I appreciate the efforts of the authors to improve paper’s quality. I have no other comments.
Author Response
Comments 1: I appreciate the efforts of the authors to improve paper’s quality. I have no other comments.
Response 1: We thank the Reviewer for their feedback which helped improve our work.